# Thalamocortical contributions to cognitive task activity

Kai Hwang[1,2,3,4]*, James M Shine[5], Michael W Cole[6], Evan Sorenson[1,2,4]

[1]Department of Psychological and Brain Sciences, University of Iowa, Iowa City, United States; [2]Cognitive Control Collaborative, University of Iowa, Iowa City, United States; [3]Iowa Neuroscience Institute, University of Iowa, Iowa City, United States; [4]Department of Psychiatry, University of Iowa, Iowa City, United States; [5]Brain and Mind Center, University of Sydney, Sydney, Australia; [6]Center for Molecular and Behavioral Neuroscience, Rutgers University-Newark, Newark, United States

**Abstract** Thalamocortical interaction is a ubiquitous functional motif in the mammalian brain. Previously (Hwang et al., 2021), we reported that lesions to network hubs in the human thalamus are associated with multi-domain behavioral impairments in language, memory, and executive functions. Here, we show how task-evoked thalamic activity is organized to support these broad cognitive abilities. We analyzed functional magnetic resonance imaging (MRI) data from human subjects that performed 127 tasks encompassing a broad range of cognitive representations. We first investigated the spatial organization of task-evoked activity and found a basis set of activity patterns evoked to support processing needs of each task. Specifically, the anterior, medial, and posterior-medial thalamus exhibit hub-like activity profiles that are suggestive of broad functional participation. These thalamic task hubs overlapped with network hubs interlinking cortical systems. To further determine the cognitive relevance of thalamic activity and thalamocortical functional connectivity, we built a data-driven thalamocortical model to test whether thalamic activity can be used to predict cortical task activity. The thalamocortical model predicted task-specific cortical activity patterns, and outperformed comparison models built on cortical, hippocampal, and striatal regions. Simulated lesions to low-dimensional, multi-task thalamic hub regions impaired task activity prediction. This simulation result was further supported by profiles of neuropsychological impairments in human patients with focal thalamic lesions. In summary, our results suggest a general organizational principle of how the human thalamocortical system supports cognitive task activity.

*For correspondence:
kai-hwang@uiowa.edu

Competing interest: The authors declare that no competing interests exist.

## Editor's evaluation

This valuable study examines a largely ignored brain structure (the thalamus) in functional brain imaging studies. The study shows that localized thalamic regions show hub properties in terms of their activation properties and connectivity to cortical regions. While some open questions regarding the robustness and validity of measure that defines the hub properties may remain, the evidence in the paper is generally convincing, especially as converging evidence across two large datasets is presented.

## Introduction

Distributed neural activity supports a broad range of perceptual, motor, affective, and cognitive functions. Discovering how brain systems implement this broad behavioral repertoire is critical for elucidating the neural basis of human cognition. Past studies have revealed two organizational principles

– low-dimensional architecture and multi-task hubs – that connect distributed neural activity with task performance across functional domains.

The application of dimensionality reduction techniques on whole-brain imaging data has revealed a relatively low-dimensional organization of cortical neural activity. Specifically, the number of variables required to explain a large amount of variance in distributed neural activity is far lower than the total number of variables in the data (*MacDowell and Buschman, 2020*; *Nakai and Nishimoto, 2020*; *Shine et al., 2019b*). The spatiotemporal patterns of these variables are commonly referred to as intrinsic networks in task-free conditions, or manifolds, motifs, and latent components in task contexts. This low-dimensional organization may reflect an elementary set of information processes implemented by distributed brain systems (*Cunningham and Yu, 2014*; *Yeo et al., 2016*), in which different tasks selectively engaged selective latent activity patterns depending on the specific processing requirements of individual tasks.

The anatomical overlap between spatiotemporal components predicts that some brain regions broadly participate in multiple tasks. In support of this prediction, studies have found that associative regions in frontal and parietal cortices are involved in executing a wide array of tasks (*Cole et al., 2013*; *Duncan, 2010*). These task-flexible regions, also commonly referred to as brain hubs (*Gratton et al., 2018*; *van den Heuvel and Sporns, 2013*), have diverging connectivity with multiple brain systems, and are thought to perform integrative functions that allow perceptual inputs to interact with contextual task representations for adaptive task control (*Bertolero et al., 2018*; *Bertolero et al., 2015*; *Ito et al., 2022*; *Nee, 2021*). The behavioral significance of brain hubs is affirmed by lesion studies demonstrating that lesions to hub regions are associated with task impairments across multiple functional domains (*Hwang et al., 2021*; *Reber et al., 2021*; *Warren et al., 2014*).

However, a prevailing assumption is that diverse human behavior depends on the organization of cortical activity, and the contribution from subcortical regions, particularly the thalamus, is not well understood. Our previous studies demonstrated that the human thalamus contains a complete representation of intrinsic cortical functional networks, and additionally, exhibits a hub-like connectivity profile interlinking multiple cortical systems (*Hwang et al., 2021*; *Hwang et al., 2017*). Furthermore, lesions to the anterior and the medial thalamus in human patients are associated with behavioral impairments across functional domains (*Hwang et al., 2020a*; *Hwang et al., 2021*). Given that every cortical region receives projections from multiple thalamic nuclei and that the thalamus mediates striatal and cerebellar influences on cortical (*Shine, 2020*), the thalamocortical system is ideally suited to shape cortex-wide activity patterns that instantiate cognitive representations. The relationship between thalamic task activity and cortical cognitive representations, however, are not well understood.

Given the observations described above, we expect that task functional magnetic resonance imaging (fMRI) data obtained from the human thalamus to exhibit a similar low-dimensional organization like the cortical system. Furthermore, there should be task hub regions within the human thalamus that broadly participate in multiple cognitive tasks. The current study applied a dimension reduction technique to determine how thalamic task-evoked activity and thalamic hubs are organized to support cognitive functions across diverse functional domains. To this end, we analyzed fMRI data where human subjects performed a rich battery of tasks designed to elicit neural activity for a wide range of cognitive functions. Our study addressed the following two specific questions. First, are there thalamic hub regions that exhibit task-evoked activity by multiple tasks across domains? Second, can thalamic task-evoked activity be used to predict cortical task-evoked activity patterns? Answering these questions would reveal a general principle of how the human thalamocortical system contributes to higher-order, multi-domain cognitive task activity.

## Results

To determine the organization of task-evoked thalamic activity, we analyzed two fMRI datasets, both of which utilized rich batteries of tasks designed to elicit a wide range of processes encompassing perceptual, affective, memory, social, motor, language, and cognitive domains (*Figure 1A*). In the first dataset, the multi-domain task battery (MDTB) dataset, 21 subjects performed 25 behavioral tasks (*King et al., 2019*). In the second dataset, the Nakai and Nishimoto (N&N) dataset, 6 subjects performed 103 tasks (*Nakai and Nishimoto, 2020*).

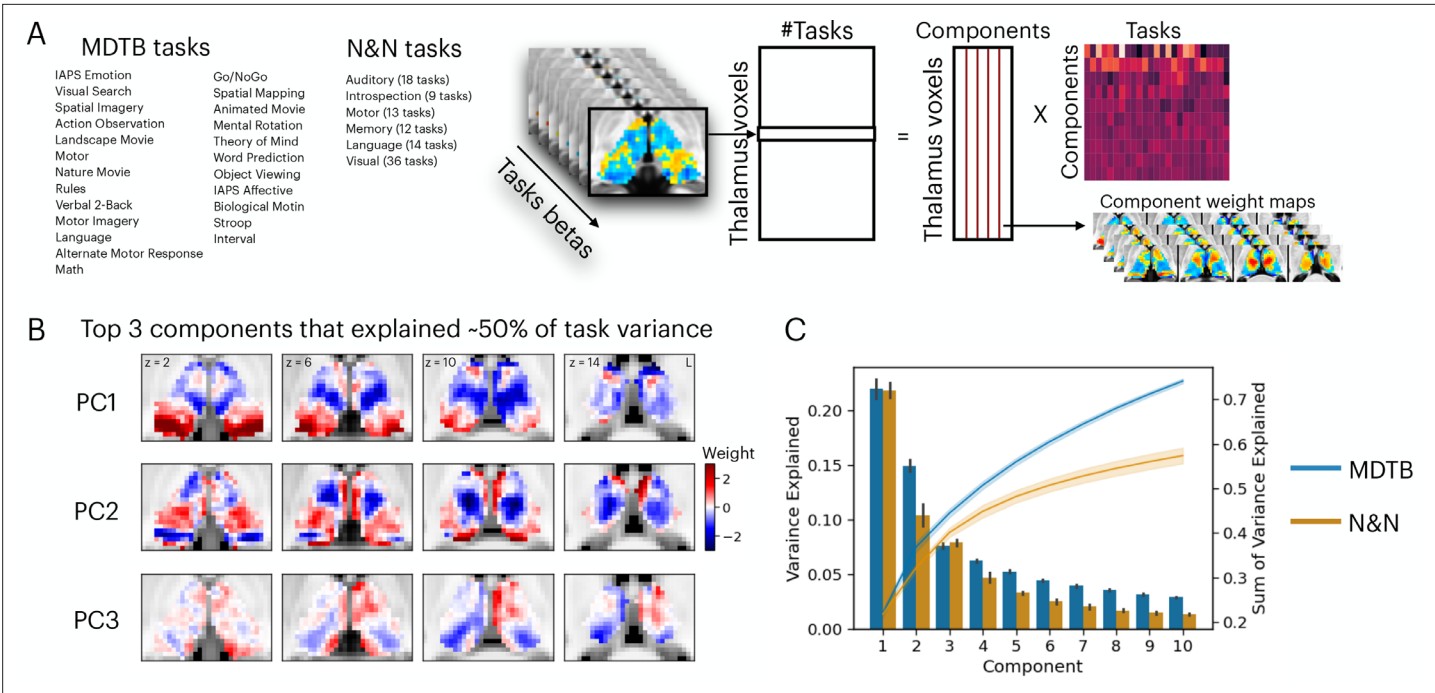

**Figure 1.** Low-dimensional organization of thalamic task-evoked response that supports multi-task performance. (**A**) We decomposed the high-dimensional multi-task evoked activity matrix into low-dimensional spatial components in the human thalamus and a task-wide loading matrix. For a list of all tasks, see ***Supplementary file 1***. (**B**) Spatial topography of the top 3 components from the multi-domain task battery (MDTB) dataset that explained 50% of the variance in the group averaged task activity matrix. For top components for the Nakai and Nishimoto (N&N) dataset, see ***Figure 1—figure supplement 1***. For the loadings between tasks and components, see ***Figure 1—figure supplement 2***. (**C**) Results from applying principal component analysis (PCA) to single subjects. For both the MDTB and N&N datasets, for individual subjects up to 57% of the variance across multiple tasks can be explained by the top 10 components. Error bars and shaded areas indicate standard error of the mean.

The online version of this article includes the following figure supplement(s) for figure 1:

**Figure supplement 1.** Spatial topography of the top three components from the Nakai and Nishimoto (N&N) dataset that explained about 50% of the variance in the group averaged task activity matrix.

**Figure supplement 2.** Loadings between individual tasks and thalamic activity components.

## Low-dimensional organization of thalamic task-evoked activity

We sought to determine the organization of task-evoked responses in the human thalamus using a dimension reduction technique. We first used a general linear modeling (GLM) approach to characterize the task-evoked blood-oxygen-level-dependent (BOLD) activity patterns and estimated the magnitude of BOLD-evoked responses for every task ('task betas'). Task betas were extracted for every thalamic voxel and every task, then compiled into a voxel-by-task activity matrix for each dataset. The cross-subject averaged matrix was subjected to a principal component analysis (PCA) to decompose multi-task BOLD activity patterns into a linear summation of a voxel-by-component weight matrix multiplied by a component-by-task loading matrix (***Figure 1A***). The voxel-by-component weight matrix can be conceptualized as sets of basis patterns of thalamic BOLD activity components engaged by different tasks. Similar to previous studies that focused on cortical activity patterns (***Nakai and Nishimoto, 2020***; ***Shine et al., 2019a***), we found that the top 3 thalamic activity components can explain up to 50% of the variance across tasks for the MDTB dataset and 36% for the N&N dataset, in which each task is associated with a weighted sum of these components (***Figure 1B***, ***Figure 1—figure supplements 1–2***). Repeating the PCA for each subject (without averaging across subjects) revealed similar findings, where up to 57% of variances across task activity patterns in the human thalamus can be explained by 10 components (***Figure 1C***; 73% for MDTB, 57% for N&N).

## Hub organization of thalamic task-evoked activity

Examining the spatial patterns of thalamic components (***Figure 1B***) revealed that several thalamic subregions showed spatial overlap across multiple components, which suggests that these thalamic

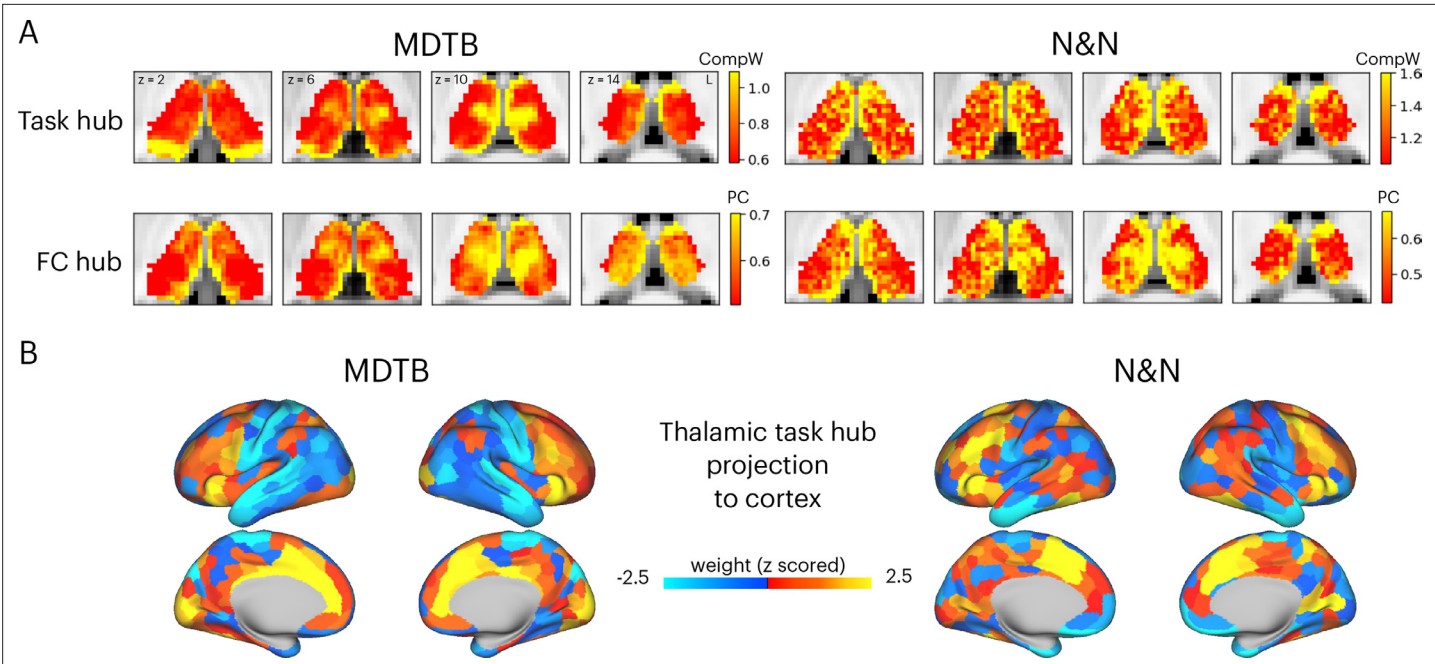

**Figure 2.** Hub regions in the thalamus. (**A**) Task hubs (CompW) and functional connectivity (FC) hubs (PC) in the thalamus. PC = participation coefficient. (**B**) Projecting task hub metrics onto the cortex via thalamocortical FC.

The online version of this article includes the following figure supplement(s) for figure 2:

**Figure supplement 1.** Control analyses for the task hub metric.

**Figure supplement 2.** Conceptual explanation of the participation coefficient metric.

subregions may support multiple tasks. We reasoned that if a thalamic voxel is broadly participating in multiple tasks, it will express stronger weights for top thalamic activity components that explained a large amount of variance in evoked activity patterns across tasks. Therefore, to further map 'task hub' regions in the human thalamus, we calculated a metric by summing each voxel's absolute component weight for the top 10 components. Results showed that for the MDTB dataset, the anterior, medial, medio-posterior, and dorsal thalamus exhibited strong task hub properties (*Figure 2A*). For the N&N dataset, the posterior thalamus did not exhibit strong task hub property, and the spatial correlation between task hub metrics from the two datasets was $r=0.36$. We then recalculated the task hub metric using lower ranked components (components number 11th to 20th, explained less than 2% of variances), and results revealed a different anatomical pattern (*Figure 2—figure supplement 1*). These results suggest that the anterior, medial, and posterior thalamic task hub patterns we identified were primarily driven by component weights summed across top components that explained large portion of variance in task-evoked activity.

Network hubs in the human thalamus exhibit diverse functional connectivity (FC) patterns inter-linking multiple cortical systems (*Greene et al., 2020*; *Hwang et al., 2017*). To determine whether the tasks hubs we identified correspond to network hub regions previously identified in the thalamus, we compared the spatial similarity of task hubs with FC hubs. We calculated a network hub metric, the participation coefficient (*Gratton et al., 2012*; *Guimerà and Nunes Amaral, 2005*; *Figure 2—figure supplement 2*), using every thalamic voxel's FC matrix with 400 cortical regions (*Schaefer et al., 2018*). When calculating FC matrices, we used timeseries after removing task-evoked variances from the preprocessed fMRI data, to reduce shared variances in task-evoked activations biasing FC estimates (see Methods for details). The task hubs showed significant spatial correspondence with FC hubs (*Figure 2A*; spatial correlation: MDTB mean = 0.17, SD = 0.083, p<0.001; N&N mean = 0.17, SD = 0.05, p<0.001; spatial correlation between MDTB and N&N: $r=0.55$). One discrepancy was the posterior thalamus, which showed strong task hub but not network hub property for the MDTB dataset. We then projected the task hubs estimates onto the cortex by calculating the dot product between each thalamic voxel's task hub estimate and its thalamocortical FC matrix. We found that

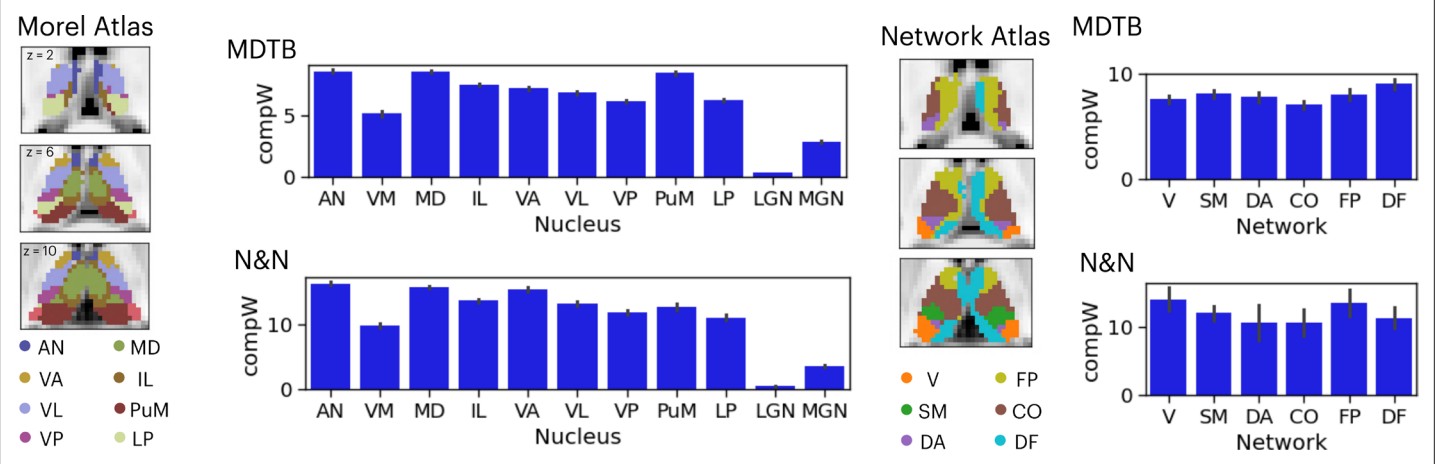

**Figure 3.** Anatomical distribution of task hub estimates in the thalamus. AN = anterior nucleus, VM = ventromedial, MD = mediodorsal, IL = intralaminar, VA = ventral anterior, VL = ventrolateral, VP = ventroposterior, PuM = medial pulvinar, LP = lateral posterior, LGN = lateral geniculate nucleus, MGN = medial geniculate nucleus; V=visual network, SM = somoatomotor, DA = dorsal attention, CO = cingulo-opercular, FP = frontoparietal, DF = default mode. Error bar indicates standard error of the mean.

tasks hubs were most strongly coupled with associate regions in the frontal and parietal cortex, for example, the lateral frontal, the insula, the dorsal medial prefrontal, and the intraparietal cortices (*Figure 2B*; spatial correlation between MDTB and N&N: *r*=0.46).

Given that the thalamus is not a homogenous structure and can be divided into different subdivisions or nuclei based on its functional and histological properties, we summarized the spatial distribution of task hub estimates (CompW) using two different thalamic atlases (*Figure 3*). First, the Morel atlas was used to determine the location of different thalamic nuclei *Krauth et al., 2010*; this atlas maps human thalamic nuclei based on cyto- and myelo-architecture information in stained slices from five postmortem human brains, and further transformed to the template space. Second, for the network atlas, we assigned each thalamic voxel to one of the seven canonical cortical function networks (*Schaefer et al., 2018*) that we used for calculating network hubs. We then averaged the task hub estimates within each thalamic nucleus or functional parcellation. In both datasets, we found the task hub values to be highest in the anterior and mediodorsal nuclei. These nuclei groups are known to have strong reciprocal connectivity with frontal regions (*Giguere and Goldman-Rakic, 1988*; *Selemon and Goldman-Rakic, 1988*). We also found the medial pulvinar to show high task hub value but only in the MDTB dataset, likely because the N&N dataset did not show high task hub value in the posterior thalamus. For the functional network parcellation atlas, between datasets discrepancies were also found in posterior thalamus. For example, in the MDTB dataset, the DF parcellation (which overlapped with the anterior medial, medial, and posterior medial thalamus) was found to show the highest task hub value. For the N&N dataset, the posterior thalamus did not exhibit high task hub value, but the FP parcellation, which covered the dorsal bank of the medial thalamus, showed the highest task hub value. Overall, these results suggest that thalamic task hubs do not overlap with one functional network parcellation or thalamic nucleus, instead overlap with multiple thalamic parcellations and nuclei.

## Predict cortical task activity with thalamic task-evoked activity

To further test whether thalamic task-evoked activity is related to diverse cognitive functions putatively implemented by distributed cortical activity, we adapted the activity flow mapping procedure (*Cole et al., 2016*; *Ito et al., 2017*) to test whether thalamic task-evoked responses can be used to predict cortical task activity (*Figure 4A*). Briefly, for every thalamic voxel, we calculated the dot product between the thalamic-evoked response pattern of each task and the thalamocortical FC matrix, using a split-half cross-validation procedure (see Methods). This calculation yielded a predicted cortex-wide activity pattern for every task, and prediction accuracy was calculated by comparing the predicted pattern to the observed cortical activity pattern using Pearson correlation. We then compared the model performance against three null models. The first null model randomly shuffled thalamic-evoked

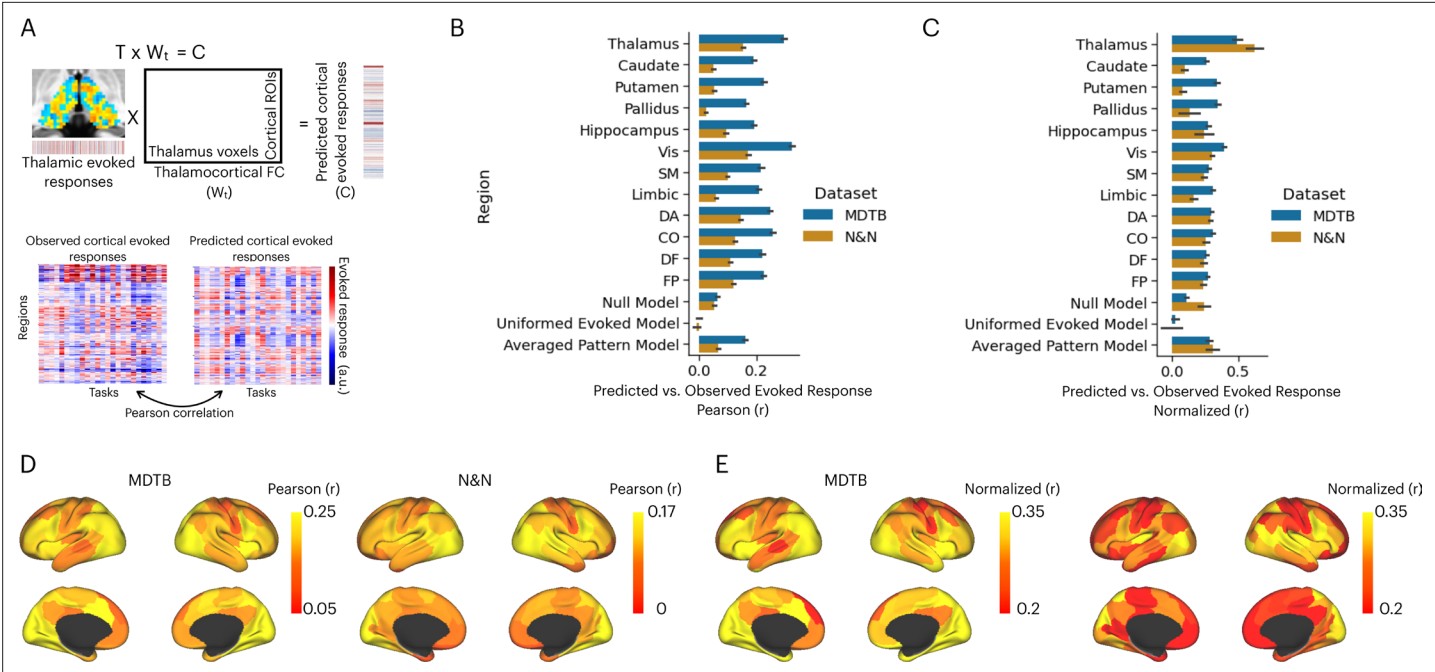

**Figure 4.** Activity flow model predicts cortical task-evoked response patterns. (**A**) Model for testing whether thalamic task-evoked activity can predict patterns of cortical task-evoked responses. (**B**) Unnormalized prediction accuracy. Cortical regions divided by seven cortical functional networks: Vis = visual, SM = somatomotor, DA = dorsal attention, CO = cingulo-opercular, DF = default mode, FP = frontal parietal; Null model: randomly permute the evoked pattern in the thalamus. (**C**) Noise normalized prediction accuracy. (**D**) Prediction accuracy normalized by the noise ceiling. (**E**) Unnormalized activity flow model prediction for 100 cortical regions of interests (ROIs). (**E**) Noise normalized activity flow model prediction for 100 cortical ROIs.

The online version of this article includes the following figure supplement(s) for figure 4:

**Figure supplement 1.** Noise ceiling of different brain regions and null models.

responses (the 'null model'), which assumed no spatial structure in thalamic task-evoked responses. The second null model set all thalamic-evoked responses to the same value ('1') across all voxels (the 'uniformed evoked model'), which assumed that cortical activity patterns are determined only by the summated inputs from thalamocortical FC patterns. The third null model was constructed by averaging the evoked response amplitudes across tasks for every thalamic voxel, which assumed that there is no difference in thalamic activity patterns between tasks. Finally, activity flow model performance is limited by the reliability of the data used to calculate the predicted pattern. Therefore, we calculated the noise ceiling that accounted for the split-half reliability of thalamic-evoked responses, thalamocortical FC, and cortical-evoked responses (see Methods for details). All prediction accuracy was normalized by the noise ceiling. For the null models, prediction accuracy was normalized by the thalamocortical model's noise ceiling.

The mean prediction accuracy of the thalamocortical activity flow model was 0.29 for the MDTB dataset and 0.15 for the N&N dataset (*Figure 4B*). As described above, the prediction accuracy is limited by the reliability of the thalamic and cortical data (noise ceiling). After accounting for the noise ceiling (*Figure 4—figure supplement 1*), the normalized prediction accuracy was 0.5 for MDTB and 0.62 for N&N (*Figure 4C*), which indicates that the thalamocortical model was able to account for up to 38% of variance in task-evoked activity patterns in the cortex. We further found that thalamocortical activity flow model outperformed the null models, except for the N&N dataset when compared to the averaged evoked pattern model (*Figure 4C*; MDTB thalamocortical model vs. null model: $t(20) = 11.22$, p<0.001; MDTB thalamocortical model vs. uniformed evoked model: $t(20) = 12.05$, p<0.001; MDTB thalamocortical model vs. averaged pattern model: $t(20) = 6.88$, p<0.001; N&N thalamocortical model vs. null model: $t(5) = 11.22$, p=0.018; N&N thalamocortical model vs. uniformed evoked model: $t(5) = 4.75$, p=0.005; N&N thalamocortical model vs. averaged pattern model: $t(5) = 1.61$, p=0.17). These results indicate that thalamic task-evoked responses can predict cortical activity patterns.

We then sought to compare our thalamocortical activity flow model to other cortical and subcortical regions. We constructed a series of comparison models by applying the same split-half cross-validation activity flow mapping procedure to 100 cortical regions of interests (ROIs) that had similar size as the thalamus (*Schaefer et al., 2018*), as well as to other subcortical regions including the hippocampus, the caudate, the putamen, and the globus pallidus. We found that several cortical regions also showed strong prediction performance, including the insula, the inferior frontal cortex, the dorsal medial frontal cortex, the intraparietal sulcus, and the lateral occipital cortex (*Figure 4D* for unnormalized prediction accuracy, *Figure 4E* for noise normalized prediction accuracy). However, the thalamocortical model outperformed all comparison models. For example for both datasets, lateral ROIs in the visual network showed the highest normalized prediction accuracy, but still statistically weaker than the thalamocortical activity flow model (MDTB: $t(20) = 2.91$, p=0.008; N&N: $t(5) = 3.3$, p=0.021). These results indicate that the thalamocortical activity flow model is a strong predictor of cortical task activity.

## Lesions to task hubs in the thalamus are associated with impaired prediction of task reorientations and representational geometry

The results presented in *Figure 4* demonstrated that thalamic task-evoked responses can predict cortical activity patterns that putatively support cognitive representations. However, it is unclear to what degree task hubs in the thalamus contribute to these results. One possibility is that all thalamic voxels contribute equally to cortical task activity patterns. Another possibility is that, since thalamic task hubs are broadly engaged by multiple tasks and exhibited diverging connectivity patterns with multiple cortical systems, thalamic task hubs have a stronger contribution to predicting cortical task activity patterns. To evaluate this prediction, we performed virtual lesion simulations. Specifically, we systematically removed 20% of thalamic voxels based on their percentile rank of task hub estimates and calculated the percentage of reduction in prediction accuracy from the activity flow analysis. For both datasets, we found that removing voxels with strong task hub estimates decreased the prediction accuracy in cortical task-evoked activity patterns (*Figure 5A*). We fitted a regression model to test the relationship between reduction in prediction and percentile rank of task hub metrics removed, and found significant negative associations (MDTB activity flow prediction: $b=-0.43$, SE = 0.019, $t=-22.63$, p<0.001; N&N activity flow prediction: $b=-0.1$, SE = 0.028, $t=-3.79$, p<0.001). These results indicate that virtual lesioning of thalamic task hubs had a stronger impact on reducing the model's ability to predict cortical activity patterns. Furthermore, thalamic voxels with the strongest impact on prediction performance were spatially located in the anterior, medial, mediodorsal, and medial posterior thalamus that we previously identified as task hubs (*Figure 5B*; spatial correlation between MDTB and N&N $r=0.29$).

We compared these simulation results with neuropsychological impairments found in 20 human patients with focal thalamic lesions. Neuropsychological profiles of these patients were described in detail in our previous publication (*Hwang et al., 2021*). Briefly, patients performed a battery of neuropsychological tests to assess executive, language, memory, learning, visuospatial, and construction functions (*Lezak et al., 2012*). Test performance was then compared to published, standardized norms and converted to $z$-scores quantify the severity of impairment (*Figure 6A*). Patients were grouped into two groups, those that showed impairment ($z$-score <−1.695, worse than 95 percentile of the normative population) in single or fewer domains (the SM group) vs. those showed impairment across multiple domains (the MM group; *Figure 6B*). There were no statistically significant differences in the lesion volumes between these two groups of patients (MM group: mean = 1559 mm$^3$, SD = 1370 mm$^3$; SM group: mean = 1073 mm$^3$, SD = 925 mm$^3$; group difference in lesion size, $t(18)$ = 0.87, p=0.39).

Lesion sites from each group were compared to the effects of the simulated lesions presented in *Figure 5*. Specifically, we compared the percentages of reduction in voxels from these two groups of lesions. We found that simulated lesions to voxels that overlapped with lesion sites in the MM patient group showed larger reductions in task activity predictions for the N&N dataset but not the MDTB dataset (*Figure 6C*; MDTB: Komogorov-Smirnov test $D=0.048$, p=0.065; N&N: Komogorov-Smirnov test $D=0.25$, p<0.001). These empirical results from human patients support our simulated lesion analyses, suggesting that lesioning task hubs in the human thalamus are associated with behavioral deficits across functional domains.

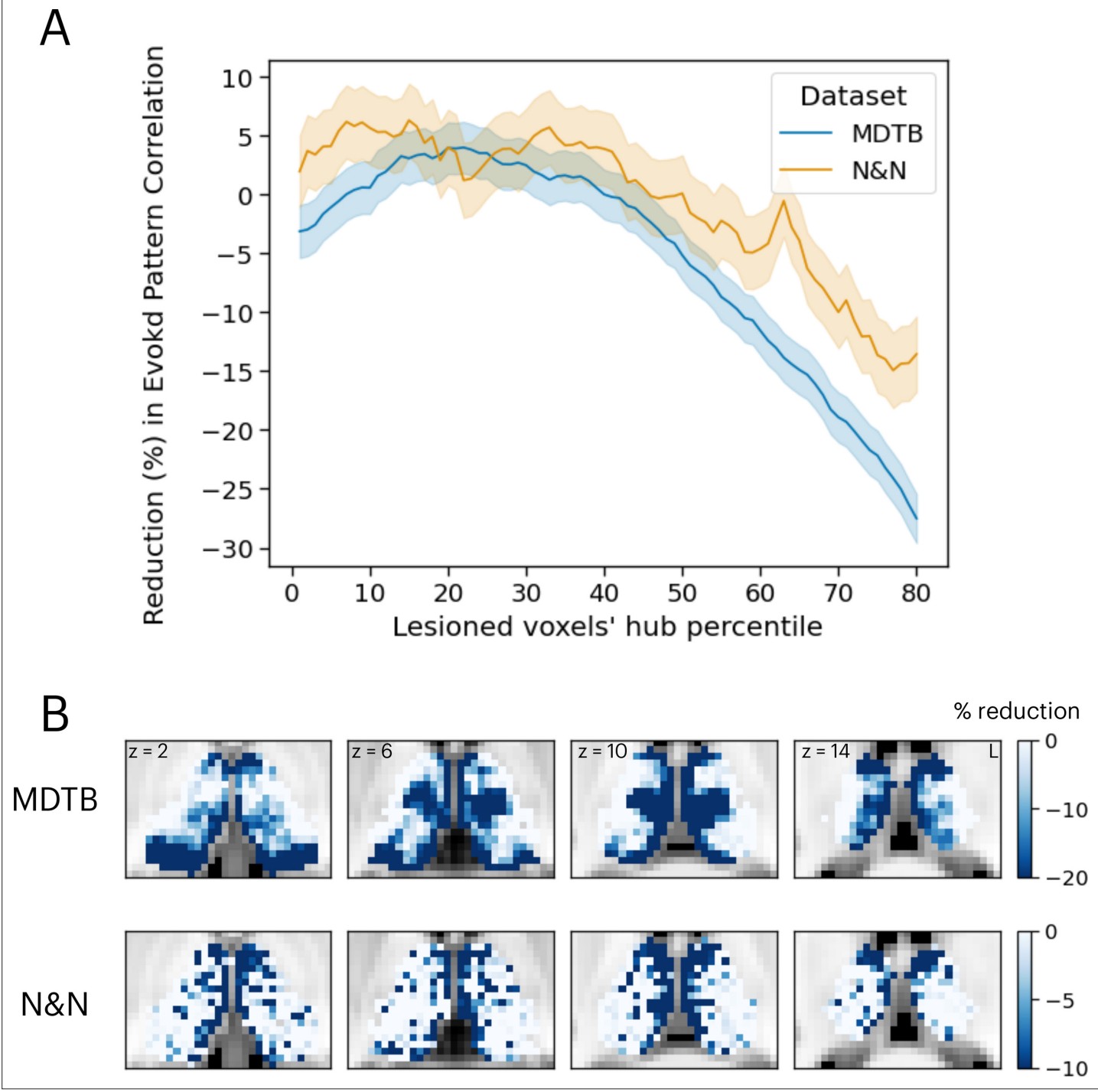

**Figure 5.** Simulating the thalamic lesion's effect on activity flow model prediction. (**A**) Artificial lesion of 20% of the thalamus voxels based on their percentile rank of task hub property: examination of the impact on cortical task-evoked activity prediction. (**B**) Subregions that showed greater reduction in prediction accuracy were primarily located in anterior, medial, and posterior thalamus. Shaded error indicates standard error of the mean.

## Discussion

It was hypothesized that the thalamus influences cognitive representations beyond sensory and motor domains (*Halassa and Sherman, 2019*; *Wolff and Vann, 2019*). Anatomically, every cortical region receives inputs from one or many thalamic subregions, and most thalamic subregions send signals to one or many cortical systems (*Jones, 2001*; *Sherman, 2007*). Functionally, resting-state fMRI studies

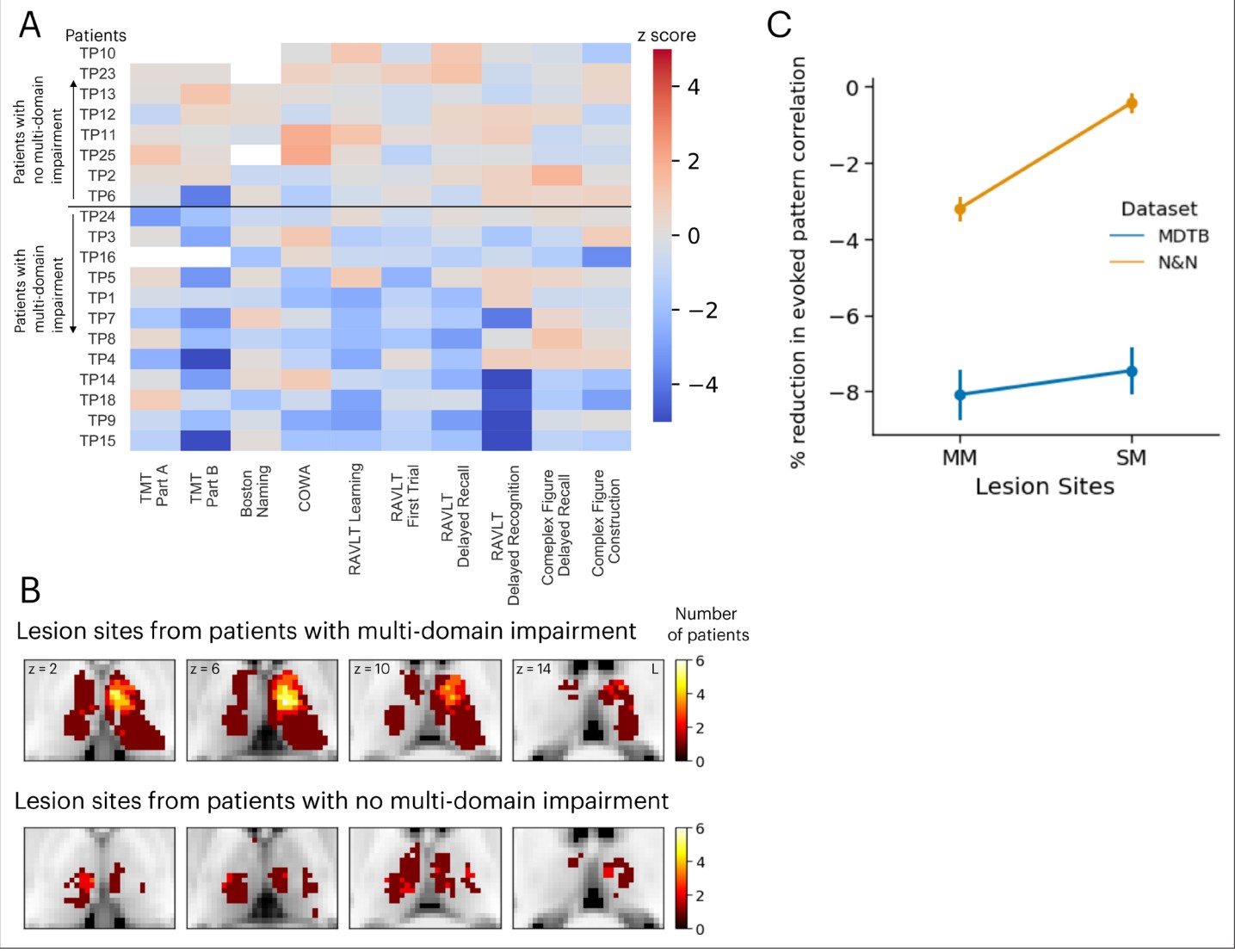

**Figure 6.** Neuropsychological evaluations from 20 patients with focal thalamic lesions. (**A**) Twelve patients exhibited multiple-domain (MM) impairment (negative *z*-scores) across multiple neuropsychological assessments. Eight patients exhibited no multi-domain (SM) impairment. (**B**) Lesion sites from patients with and without multi-domain impairment. (**C**) Mean and standard error of the mean of the reduction in activity flow model prediction after virtual lesions, plotted separately for virtual lesion sites that overlapped with MM or SM lesions. Error bar indicates standard error of the mean. Panel A reproduced from Figure 2B from *Hwang et al., 2021*, with permission.

have found a complete representation of intrinsic cortical networks in the human thalamus (*Greene et al., 2020*; *Hwang et al., 2017*; *Yuan et al., 2016*; *Zhang et al., 2008*). Several of these networks, such as the FP, CO, and DF, have been implicated in multiple cognitive functions (*Sadaghiani et al., 2010*; *Seeley et al., 2007*). These observations raised a broader question of if and how the human thalamus contributes to cognitive task activity across functional domains. The current study addressed this question by investigating how thalamic task-evoked activity are organized and how it can predict cortical cognitive task activity.

We identified several key organizational characteristics. First, the anterior, medial, and posterior-medial thalamus participated in multiple tasks, and these 'task hubs' spatially overlapped with FC hubs mapped in our previous studies (*Hwang et al., 2021*; *Hwang et al., 2017*). Second, thalamic task-evoked activity can predict task-specific activity patterns in the cortex via a linear thalamocortical activity flow model. Critically, when compared to comparison models and models built on other brain structures, this thalamocortical activity flow model performed the best in predicting cortical task activity, highlighting the capacity of the thalamus in influencing distributed activity patterns

that may instantiate cognitive representations. Finally, findings from the thalamocortical activity flow model were further corroborated by simulated lesions and neuropsychological impairments observed in patients with focal thalamic lesions, affirming the behavioral significance of thalamocortical interactions. Collectively, these findings highlight a general and critical role the human thalamocortical system supporting cognitive functions.

Neural systems support a rich and diverse behavioral repertoire. Human functional neuroimaging and studies of animal models found a low-dimensional organization scheme in cortical neural activity (*Beam et al., 2021*; *Karolis et al., 2019*; *MacDowell and Buschman, 2020*; *Nakai and Nishimoto, 2020*; *Shine et al., 2019b*). Given this finding, multi-task fMRI data obtained from the human thalamus should exhibit similar property. In our study, we applied a dimension reduction procedure to decompose multi-task thalamic activity into low-dimensional task components. Notably, we found thalamic task components that explained large amount of variances across tasks expressed strongest weights in anterior, medial, and posterior thalamus, but not in lateral thalamic regions that overlap with first-order thalamic relays such as the lateral geniculate nucleus and the ventrolateral nucleus.

Because of its unique connectivity profile, the thalamus may be in an ideal position to influence cortex-wide task activity and the diverse cognitive functions cortical activity patterns instantiate. Because every cortical region receives inputs from one or multiple thalamic regions, thalamocortical connectivity may be more effective in pushing cortical activity patterns to the desired task state. We speculate that the thalamus may be at the nexus where specific activity patterns in the thalamus can be selectively down- or up-regulated to meet the specific information processing demands of different tasks. Functions that may be influenced by the thalamocortical system suggested by previous studies include selection and gain control (*Halassa and Kastner, 2017*), adjustment of inter-regional communication (*Hwang et al., 2020b*; *Saalmann et al., 2012*), and modulation of cortical excitability (*Kosciessa et al., 2021*). The small size of the thalamus further suggests a potential role in modulating the dimensionality of task-evoked cortical activity – for instance, previous studies have found that thalamic activity correlates with the dimensionality and the strength of cross-system coupling between cortical regions (*Garrett et al., 2018*; *Shine et al., 2019b*), and thalamic lesions disrupt the low-dimensional cortical network organization (*Hwang et al., 2017*).

The studies described above suggest a strong relationship between thalamic activity and cortical cognitive activity. We further tested whether we can predict cortical activity patterns based on thalamic task-evoked activity and thalamocortical FC pattern with cortical systems. We found that our data-driven thalamocortical activity flow model can indeed successfully predict task-specific activity patterns observed in the cortex, better than null models that assumed that the thalamocortical system does not carry task-specific information. While the observed prediction accuracy was lower than previous studies that focused on cortico-cortical activity flow models (*Cole et al., 2016*), most of those studies included multiple cortical regions to build the prediction model. Our study tested a different question to evaluate the predictive power of a single region, the thalamus, relative to other brain regions. We found that after accounting for the noise ceiling, thalamic activity alone was able to predict approximately 38% of the systematic variance of cortical activity patterns across tasks. Critically, we found that the thalamocortical model outperformed comparison models constructed with other cortical and subcortical structures, including frontal (the insula and the inferior frontal sulcus), parietal, and striatal regions previously implicated in adaptive control of tasks (*Badre and Nee, 2018*; *Ito et al., 2022*). This result suggests that the thalamocortical system exerts a strong influence on cortical activity that instantiates cognitive representations.

Placing the thalamus in a central position for influencing cortex-wide cognitive activity has several benefits. For instance, the thalamus consists of two classes of thalamocortical projection cells, 'matrix' and 'core' cells, where core cells target specific granular cortical structures, and matrix cells diffusely project to multiple brain regions (*Jones, 2001*). Given that different brain regions may be encoding different perceptual and cognitive information (*Christophel et al., 2017*), selectively modulating thalamic activity may facilitate targeted activations of cortical representations via associated thalamocortical interactions. The diffuse projection pattern of matrix cells in the thalamus may simultaneously activate multiple cortical regions and promote inter-system integration when required (*Hwang and D'Esposito, 2022*; *Jones, 2001*). Furthermore, the thalamus is modulated by inputs from other subcortical structures, such as the reticular nucleus, the basal ganglia, the superior colliculus, and the cerebellum (*Bostan and Strick, 2018*; *Hwang et al., 2020b*). This anatomical feature suggests that

the thalamus may be an efficient target for exerting neuromodulatory (e.g., dopaminergic) influences on cognitive processes when required (*Castro-Alamancos and Gulati, 2014*; *Garrett et al., 2022*). Several computation models have leveraged these characteristics to explain the functions of basal-ganglia-thalamic circuits for higher-order cognitive functions (*Jaramillo et al., 2019*; *O'Reilly and Frank, 2006*).

In our previously study (*Hwang et al., 2017*), we found that the anterior, medial, and posterior thalamus to have strong network hub properties, exhibiting strong FC with multiple cortical systems in frontal and parietal cortices. Network hubs in the frontal and parietal cortices have also been found to broadly participate in cognitive tasks across different functional domains (*Bertolero et al., 2015*; *Yeo et al., 2016*), likely reflect its role in instantiating task representations (*Schumacher and Hazeltine, 2016*) and support cross-domain cognitive functions that arise from brain-wide network interactions (*Gratton et al., 2018*; *van den Heuvel and Sporns, 2013*). We hypothesized that network hubs in the thalamus may share this property with cortical network hubs. However, our previous study utilized resting-state FC, and multi-task fMRI data are needed to test task participation. In the current study, we analyzed task-evoked activity patterns in the thalamus and found task hub regions overlapped with network hubs in the anterior, medial, and posterior thalamic regions.

We found that thalamic task hubs were most strongly coupled with multiple regions across frontal and parietal associative cortices, including subregions that found to exhibit functional specialization for specific tasks as well as task-flexible regions (*Yeo et al., 2016*). Several selective regions within the frontal and parietal cortices have been hypothesized to be part of a flexible multiple-demand system (*Assem et al., 2020*) and thus, these thalamic subregions may be part of a core system whose functions are required to perform specific computations that are common across many different task contexts. One strong test of the behavioral relevance of these thalamic task hubs is to examine the effects of lesions. If thalamic task hubs are part of a domain-general core, lesioning task hubs should be associated with broad behavioral deficits not confined to a specific behavioral domain. Indeed, we found that simulated lesions to thalamic task hubs impaired prediction accuracy in cortical activity pattern across tasks, and stroke lesions to the anterior-medial thalamus in human patients were associated with behavioral deficits across functional. At this point we are agnostic to the specific function these thalamic task hubs perform. One hypothesis, suggested by findings from rodent models, is that the medial thalamus excites task-relevant and inhibits task-irrelevant cortical activity that encodes working memory representations (*Mukherjee et al., 2021*; *Rikhye et al., 2018*). Thus, in human subjects, thalamocortical interactions between thalamic task hubs and frontoparietal systems may control task representations that bind sensory, motor, and contextual information that are necessary for adaptive behavior across tasks (*Schumacher and Hazeltine, 2016*). This is an open question that should be evaluated in future studies.

On important limitation of our study is that our thalamocortical activity flow model cannot establish the directionality of thalamic activity influencing cortical activity. It is more likely that brain systems instantiate cognitive representations via recurrent connectivity among cortical regions and the thalamus, as well as integrating inputs from other brain structures. As discussed above, including all brain regions into the activity flow model can substantially improve model performance in predicting task-specific activity patterns (*Cole et al., 2016*). Nevertheless, our findings suggest that, across the whole brain, the thalamus likely has one of the strongest contributions to cortical task activity. Another important limitation is that the posterior thalamus exhibited strong task hub property in the MDTB dataset but not the N&N dataset. We suspect this could be related to differences in behavioral tasks across the two datasets, and speculatively, to the increased noise in the N&N dataset, given the lower number of observations per task and lower subcortical signal-to-noise ratio. One strong test of mapping task hubs is the lesion methods, and in our previous study (*Hwang et al., 2021*) we did not have good lesion coverage on the pulvinar nucleus. Future studies should test whether patients with pulvinar lesions have cognitive deficits beyond visual attention tasks (*Snow et al., 2009*). To conclude, our study reported important organizational principles of how thalamic task-evoked activity and thalamocortical connectivity supports cognitive tasks. These results highlight how the thalamus is in a central position for supporting cognitive task representations.

## Methods

### Datasets

We analyzed two datasets, the MDTB dataset (*King et al., 2019* at https://openneuro.org/datasets/ds002105/) and the N&N dataset (*Nakai and Nishimoto, 2020* at https://openneuro.org/datasets/ds002306/). We selected these datasets because both include fMRI data from subjects that performed a large number of tasks across functional domains (*Figure 1A*). The MDTB dataset included fMRI data from 21 subjects (13 women, 8 men; mean age = 23.28 years, SD = 2.13; we excluded 3 out of the original 24 subjects that did not have all tasks available after removing high noise datapoints). MRI data were collected on a 3T Siemens Prisma with the following parameters: repetition time (TR)=1 s; 48 slices with 3 mm thickness; in-plane resolution 2.5×2.5 mm$^2$; multi-band acceleration factor = 3; in-plane acceleration factor = 2; echo time and flip angle were not reported in the original paper. Structural T1 images were acquired using magnetization-prepared rapid acquisition gradient echo sequence (MPRAGE; field-of-view [FOV]=15.6 × 24 × 24 cm$^3$, at 1 × 1 × 1 mm$^3$ voxel size). The N&N dataset included fMRI data from six subjects (two females, four males, age range 22–33 years) collected on a 3T Siemens Tim Trio system. fMRI data were collected with the following parameters: TR = 2 s; 72 slices with 2 mm slice thickness; in-plane resolution = 2 × 2 mm$^2$; echo time = 30 ms; flip angle = 62 degrees; multi-band factor = 3. Structural MRI data were collected with an MPRAGE sequence (TR = 2530 ms; TE = 3.26 ms; flip angle = 9 degrees; FOV = 256 × 256 mm$^2$; voxel size = 1 × 1 × 1 mm$^3$).

### MRI data preprocessing

Both datasets were preprocessed using fMRIPrep version 20.1.1 (*Esteban et al., 2019*) to reduce noise and transform data from subject native space to the ICBM 152 Nonlinear Asymmetrical template version 2009c for group analysis (*Fonov et al., 2011*). Preprocessing steps include bias field correction, skull-striping, co-registration between functional and structural images, tissue segmentation, motion correction, and spatial normalization to the standard space. As part of preprocessing, we obtained nuisance regressors including rigid-body motion estimates, cerebral spinal fluid (CSF), and white matter (WM) noise components from the component-based noise correction procedure (*Behzadi et al., 2007*). These nuisance regressors were entered in the regression model to reduce the influences from noise and artifacts (see below). We did not perform any spatial smoothing.

### Task-evoked responses

The MDTB dataset contained 24 cognitive tasks and a resting condition collected across four scanning sessions; each session consisted of eight runs. Each task started with a 5 s instruction and was followed by 30 s of continuous task performance. The N&N dataset contained 103 tasks collected across 18 MRI scanning runs in 3 days. Each task trial lasted 6–12 s where task instruction was on screen throughout the trial, and each task repeated 12 times distributed across runs. For a complete list of behavioral tasks, see *Figure 1A* and *Figure 1—figure supplement 1*. The detailed design of each task for both datasets was reported in *King et al., 2019*, and in *Nakai and Nishimoto, 2020*.

We employed a voxel-wise GLM approach to estimate task-evoked responses, using AFNI's 3dDeconvolve (*Cox, 1996*). For every voxel, a generalized least squares regression model was constructed and fitted to the preprocessed timeseries with the following regressors: task regressors, rigid body motion regressors and their derivatives, and the top 5 CSF and WM noise components. High motion volumes (framewise displacement >0.2 mm) were removed from data analyses via the censoring option, and subjects with tasks with more than 40% of the data censored were dropped from further analyses (three MDTB subjects dropped). For each subject, all imaging runs were concatenated, and signal drifts were modeled separately for each run, including run-specific constant and polynomial regressors. All task regressors were created by convolving a gamma hemodynamic response function with the stimulus duration. The stimulus duration was determined by the specific block or trial design sequence of each task (see *Figure 1—figure supplement 1* for the specific duration used for all tasks). For the MDTB dataset, several tasks contained different sub-conditions, and the GLM estimates of sub-conditions were averaged to obtain one estimate per task. Repeating our analyses without averaging sub-conditions did not change our results. For the MDTB dataset, the 5 s instruction period at the start of each task block was modeled but not included into subsequent analyses. For the N&N dataset, the task instruction was presented simultaneously during all trials, and

as a result it could not be separated from the GLM analyses. The residuals from this GLM were saved for subsequent FC analysis.

## Low-dimensional organization of task-evoked activity in the thalamus

To probe the of task-evoked activity in the thalamus, we first applied a thalamus mask from the Harvard-Oxford subcortical atlas, which mask was used to extract task-specific evoked response estimates from 2445 thalamus voxels. These estimates were then compiled into a task-by-voxel evoked activity matrix for each subject. This evoked activity matrix was then *z*-scored by subtracting the grand mean from the whole matrix and divided by the standard deviation across all elements. A PCA decomposed the evoked activity matrix into a linear summation of a voxel-by-component weight matrix multiplied by a component-by-task loading matrix (*Figure 1A*). The voxel-by-component weight matrix can be conceptualized as sets of basis patterns of thalamic activity components engaged by different tasks. The loading matrix described the relationship between each voxel-wise component map and tasks. We then summated the percentage of variances explained to determine how many components were required to explain more than 50% of the variance observed in the evoked activity matrix. We performed the PCA with and without averaging the evoked activity matrix across subjects. We averaged the matrix before PCA to visualize the voxel patterns of each component and repeated the PCA without averaging separately for each subject to test whether a low-dimensional organization can be replicated at the level of individual subjects.

## Thalamic task hubs

Some thalamic subregions could participate in multiple tasks across functional domains, exhibiting 'task hub' properties. To map these task hub regions, for every voxel we calculated a *CompW* metric:

$$CompW = \sum_{i=1}^{n} |W_i|$$

where $|W_i|$ is the absolute value of weight for component *i*, and *n* is the number of components. For the main analysis, we calculated this metric for the top 10 components. For the supplemental analysis (*Figure 2—figure supplement 1*), we calculated *CompW* for the 10th to the 20th component. We reasoned that a task hub region would be more strongly recruited by the top 10 components that explained a large amount of the variances in the evoked activity matrix, exhibiting a higher *CompW* estimate.

To compare the task hubs to the FC hubs that we had mapped previously (*Hwang et al., 2017*), we first obtained a thalamocortical FC matrix using principal component linear regression (*Ito et al., 2017*) to estimate patterns of FC between each thalamic voxel and 400 cortical ROIs (*Schaefer et al., 2018*). We estimated this FC matrix using residuals after task regression. Note that one advantage of principle component linear regression is that its estimates are similar to partial correlation, accounting for shared variances among signals. We then calculated a connector hub metric participation coefficient for each voxel (*Gratton et al., 2012*):

$$PC = 1 - \sum_{s=1}^{N} \left(\frac{K_{is}}{K_i}\right)^2$$

where $K_i$ is the sum of total FC weight between voxel *i* and all cortical ROIs, $K_{is}$ is the sum of FC weight between voxel *i* and cortical ROIs in cortical network *s*, and *N* is the total number of cortical networks. If a voxel has connectivity uniformly distributed to all cortical networks, then its PC value will be close to 1; otherwise, if its connectivity is concentrated within a specific cortical network, its PC value will be close to 0. To perform this calculation, we assigned the 400 cortical ROIs to seven cortical functional networks, that is, to the FP, DF, CO, DA, limbic, SM, and visual networks (*Schaefer et al., 2018*; *Yeo et al., 2011*). We calculated participation coefficient values across a range of density thresholds of the thalamocortical FC matrix (density = 0.01–0.15) and averaged across thresholds.

To determine which cortical regions showed the strongest coupling with thalamic task hubs, we calculated the dot product between *CompW* and thalamocortical FC matrix, yielding a project task hub matrix on the cortical space:

$$Cortical\ CompW = CompW \cdot FC_{ThxCtx}$$

where *CompW* is the 2445 voxel-wise task hub metrics, and $FC_{ThxCtx}$ is the 2445×400 thalamocortical FC matrix.

## Thalamocortical activity flow mapping

To determine whether thalamic task-evoked activity can predict cortical task activity patterns, we modified the activity flow mapping procedure (*Cole et al., 2016*; *Ito et al., 2017*), and for each task and each subject we calculated:

$$A_{ctx} = B_t \cdot FC_{ThxCtx}$$

where $B_t$ is the evoked response estimate for every thalamic voxel, $FC_{ThxCtx}$ is the thalamocortical FC matrix, and $A_{ctx}$ is the predicted cortical activity pattern. $A_{ctx}$ was calculated with a split-half cross-validation procedure. For both datasets, we split the data into two halves to calculate two sets of $B_t$ , $FC_{ThxCtx}$ , and $A_{ctx}$ estimates. $A_{ctx}$ was then empirically compared to the observed activity patterns from the other half of the data (cross-validation), using Pearson correlation. Pearson correlation was calculated across cortical ROIs separately for each task, then averaged across tasks. The correlation values were then averaged across the two split-half estimates. For calculating $FC_{ThxCtx}$ , we used principal component linear regression (as described above) to estimate patterns of FC between timeseries. Critically, we used residuals from the GLM instead of the preprocessed functional timeseries to minimize the potential confound of averaged evoked responses inflating estimates of FC (*Cole et al., 2019*). To verify that task-evoked activations have been successfully removed from the timeseries, we correlated the task regressor (stimulus timing convolved with the gamma response function) with the thalamus and cortical timeseries that we used for calculating $FC_{ThxCtx}$ . We found the correlations to be weak (with cortical ROI timeseries, mean $r=-2.02 * 10^{-10}$, SD = $1.06* 10^{-8}$; with thalamic voxel timeseries, mean $r=4.37 * 10^{-10}$, SD = $1.27* 10^{-10}$). These weak correlations suggest that task-evoked activations likely did not bias our activity flow analysis.

To evaluate our thalamocortical activity flow mapping procedure, we compared its results to three different null models and to comparison models. The first null model randomly shuffled the voxel order in $W_t$ , which assumed no task-specific information in thalamic-evoked activity pattern. The second null model set $W_t$ to a uniform value (1 for every voxel), which assumed that there is no spatial variance in the thalamic-evoked response pattern and thus $A_{ctx}$ was entirely determined by the summed of connectivity weights in $FC_{ThxCtx}$ . The third null model was constructed by calculating an averaged estimate across all conditions for every voxel, which assumed there is no difference in evoked response patterns between conditions. We also constructed a series of comparison models by repeating the same analysis on other source ROIs, including the hippocampus, the caudate, the putamen, and the pallidus, and 100 cortical ROIs (*Schaefer et al., 2018*) that were similar in size to the thalamus. For these comparison models, we replaced $W_t$ with voxel-wise evoked responses from other source ROIs, and recalculated the FC matrix by calculating FC between every voxel in the source ROI and 400 cortical ROIs using PCA linear regression.

To evaluate the activity flow model, it is important to interpret model performance relative to the noise ceiling of the data (i.e., the reliability), which is limited by the reliability of the source ROI, the FC matrix, and cortical task-evoked activity patterns. Thus, following the procedures developed by *King et al., 2022*, we estimated the noise ceiling of every model, task, and source ROI with the following:

$$NoiseCeiling = \sqrt{Reliability\left(Modeled\_A_{ctx}\right) \times Reliability\left(Observed\_A_{ctx}\right)}$$

where $Reliability\left(Modeled\_A_{ctx}\right)$ is the split half reliability of $A_{ctx}$ of the source ROI, and $Observed\_A_{ctx}$ is the split half reliability of the observed cortical-evoked responses. We quantified the noise ceiling of each model, source ROI, task, and subject, and then normalized the activity flow prediction accuracy by dividing $A_{ctx}$ by the estimated noise ceiling.

## Lesion analyses

We performed additional simulated and empirical lesion analyses to test which thalamic subregions contribute to predicting cortical task activity. For simulated lesions, we ranked thalamic voxels by their task hub (*CompW*) metrics, then set the evoked responses in 20% of thalamic voxels to zero and repeated the thalamocortical activity flow analysis. We repeated this virtual lesioning procedure

for voxels ranking from 0% to 80% in steps of 1%, window size of 20%, and calculated the effects of virtual lesioning of thalamic voxels on predicting individual task activity pattern. The percentage of prediction reduction (relative to the original observed value) after virtual lesioning was assigned to each voxel to construct voxel-wise maps that depict the effects of simulated lesions.

We then compared the effects of simulated lesions to lesions observed in human patients with focal thalamic lesions. Details of these patients were reported in our previous study (*Hwang et al., 2021*). Briefly, these patients were selected from the Iowa Neurological Patient Registry, and had focal lesions caused by ischemic or hemorrhagic strokes restricted to the thalamus (age = 18–70 years, mean = 55.8 years, SD = 13.94 years, 13 males). The lesion sites of these patients were manually traced and normalized to the MNI-152 template using a high-deformation, nonlinear, enantiomorphic, registration procedure that we described in detail in our previous papers (*Hwang et al., 2020a*; *Hwang et al., 2021*). All participants gave written informed consent, and the study was approved by the University of Iowa Institutional Review Board (IRB protocol #200105018).

We further tested whether voxels associated with stronger virtual lesioning effects overlapped with lesion sites associated with more pronounced behavioral deficits in human stroke patients. Behavioral deficits were assessed using a set of standardized neuropsychological tests: (1) executive function using the Trail Making Test Part B (TMT Part B); (2) verbal naming using the Boston Naming Test (BNT); (3) verbal fluency using the Controlled Oral Word Association Test (COWA); (4) immediate learning using the first trial test score from the Rey Auditory-Verbal Learning Test (RAVLT); (5) total learning by summing scores from RAVLT, trials 1 through 5; (6) long-term memory recall using the RAVLT 30 min delayed recall score; (7) long-term memory recognition using the RAVLT 30 min delayed recognition score; (8) visuospatial memory using the Rey Complex Figure delayed recall score; (9) psychomotor function using the Trail Making Test Part A (TMT Part A); and (10) construction using the Rey Complex Figure copy test. To account for age-related effects, all test scores were converted to age-adjusted $z$-scores using the mean and standard deviation from published population normative data. We determined the functional domain that each test assessed, described in Neuropsychological Assessment (*Lezak et al., 2012*). Twelve out of 20 patients had significant impairment ($z < -1.645$) reported in more than two functional domains, and thus were classified as the MM patient group. The rest of the 8 patients had significant impairment in one or fewer domains, and classified as the SM group. We predicted that lesion sites in the MM group would show stronger virtual lesioning effects, when compared to lesion sites in the SM group.

## Code and data availability statement

Code and data are available at https://github.com/HwangLabNeuroCogDynamics/ThalamicTaskHubs; (copy archived at swh:1:rev:d876ea93f1174ffdfe35c60f1f8b0ba8b637b037; *Hwang, 2022*).

## Acknowledgements

ES and KH were supported by National Institute of Mental Health R01MH122613. The content is solely the responsibility of the authors and does not represent the official views of the National Institutes of Health.

## Additional information

### Funding

| Funder | Grant reference number | Author |
| --- | --- | --- |
| National Institute of Mental Health | R01MH122613 | Kai Hwang |

The funders had no role in study design, data collection and interpretation, or the decision to submit the work for publication.

## Author contributions
Kai Hwang, Conceptualization, Resources, Software, Formal analysis, Supervision, Funding acquisition, Validation, Investigation, Visualization, Methodology, Writing – original draft, Project administration, Writing – review and editing; James M Shine, Conceptualization, Methodology, Writing – review and editing; Michael W Cole, Software, Methodology, Writing – review and editing; Evan Sorenson, Resources, Data curation, Software, Formal analysis, Investigation, Visualization, Methodology, Writing – review and editing

## Author ORCIDs
Kai Hwang (ID) http://orcid.org/0000-0002-1064-7815
James M Shine (ID) http://orcid.org/0000-0003-1762-5499
Michael W Cole (ID) http://orcid.org/0000-0003-4329-438X

## Ethics
All participants gave written informed consent, and the study was approved by the University of Iowa Institutional Review Board (IRB protocol #200105018).

## Decision letter and Author response
Decision letter https://doi.org/10.7554/eLife.81282.sa1
Author response https://doi.org/10.7554/eLife.81282.sa2

# Additional files

## Supplementary files
- Supplementary file 1. List of task conditions.
- MDAR checklist

## Data availability
Raw data are available at OpenNeuro.org (https://openneuro.org/datasets/ds002105/ and https://openneuro.org/datasets/ds002306/). Code and data are available at (https://github.com/HwangLabNeuroCogDynamics/ThalamicTaskHubs; copy archived at swh:1:rev:d876ea93f1174ffdfe35c60f1f8b0ba8b637b037).

The following previously published datasets were used:

| Author(s) | Year | Dataset title | Dataset URL | Database and Identifier |
| --- | --- | --- | --- | --- |
| King M, Hernandez-Castillo CR, Poldrack RA, Ivry RB, Diedrichsen J | 2019 | multi-domain task battery | https://openneuro.org/datasets/ds002105/versions/1.1.0 | OpenNeuro, 10.18112/openneuro.ds002105.v1.1.0 |
| Nakai T, Nishimoto S | 2020 | Over 100 Task fMRI Dataset | https://openneuro.org/datasets/ds002306/versions/1.1.0 | OpenNeuro, 10.18112/openneuro.ds002306.v1.0.3 |

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
