## [Editor Report]

This valuable study examines a largely ignored brain structure (the thalamus) in functional brain imaging studies. The study shows that localized thalamic regions show hub properties in terms of their activation properties and connectivity to cortical regions. While some open questions regarding the robustness and validity of measure that defines the hub properties may remain, the evidence in the paper is generally convincing, especially as converging evidence across two large datasets is presented.

---

## [Decision Letter]

**Decision letter after peer review:**

Thank you for submitting your article "Thalamocortical contribution to cognitive task activity" for consideration by *eLife*. Your article has been reviewed by 2 peer reviewers, including Jörn Diedrichsen as Reviewing Editor and Reviewer #1, and the evaluation has been overseen by Timothy Behrens as the Senior Editor. The following individual involved in the review of your submission has agreed to reveal their identity: Moataz Assem (Reviewer #2).

Essential revisions:

1. The measure of "task hub" properties that is central to the paper would need to be much better explained and justified. You motivate the measure to be designed to find voxels that are "more flexibly recruited by multiple thalamic activity components", but it is not clear to me at this point that the measure defined on line 634 does this. First, sum_n w_i^2^ is constrained to be the variance of the voxel across tasks, correct? Would sum_n abs(w) be higher when the weights are distributed across components? Given that each w is weighted by the variance (eigenvalue) of the component across the thalamus, would the score not be maximal if the voxel only loaded on the most important eigenvector, rather than being involved in a number of components? Also, the measure is clearly not rotational invariant – so would this result change after some rotation PCA solution? Some toy examples and further demonstrations that show why this measure makes sense (and what it really captures) would be essential. The same holds for the participation index for the resting state analysis.

2. The least compelling set of results (though not necessarily wrong) is the thalamic prediction of cortical activations. This is because the functional connectivity (FC) matrix used to link the thalamus and cortex was derived from the same data after regressing out task-related variance. However, this process might not be clean enough. The authors do not provide enough details on how task-evoked responses were regressed and how the residuals were assessed to be clean. A stronger test would utilize an FC matrix derived from independent data. Alternatively, I suggest using the FC matrix from dataset 2 to examine cortical projections of dataset 1 (and vice versa).

3. Throughout the manuscript, there is a general dependence on qualitative comparisons instead of quantifying similarities between findings. For example, (a) the spatial similarity of task hubs across the two datasets was not assessed (Figure 1d) (b) similarities between thalamic task hub projections to the cortex (Figure 2b). The comparison between the two datasets should be better quantified.

4. For the activity flow analysis, the null models (which need to be explained better) appear weak (i.e. no differences across tasks?), and it is no small wonder that the thalamus does significantly better. The Pearson correlations are not overwhelmingly impressive either. To give the reader a feel for how good/bad the prediction actually is, it would be essential that the authors would report noise ceilings – i.e. based on the reliability of the cortical activity patterns and thalamic activity patterns, what correlation would the best model achieve (see King et al., 2022, BioRxiv, as an example).

*Reviewer #2 (Recommendations for the authors):*

The findings presented here are important. The following recommendations would make the conclusions stronger:

1. The study would benefit from linking the thalamic task hubs to canonical resting-state networks defined in the thalamus (e.g. Ji et al. 2019 NeuroImage). Do task hubs mainly overlap with one functional network [e.g. the frontoparietal network (FPN)] or do they cross multiple functionally distinct networks? The latter would suggest some fractionation of the identified hubs (which is alluded to in the current results, due to the non-replication of the posterior thalamic task hub across datasets). It would be most interesting if task hubs ended up occupying voxels at the intersection of multiple RSNs (similar to how Power et al. 2013 Neuron defined hubs).

2. Could fractionating the thalamic task hubs reveal different cortical contributions? The cortical results in Figure 2B are actually not that similar across the datasets (e.g. ventro-medial frontal and posterior cingulate areas). Further, the cortical areas identified occupy much of the association cortices, which is inconsistent with more localized cortical hubs that the authors reference (Bertolero et al. as well as other studies). The posterior portion of the thalamic task hubs is already one potential contributing factor to these differences. There is also evidence that anterior thalamic regions are the most connected to a localized core subset of the FPN (Assem et al. 2020 Cerebral Cortex). Similarly, would simulating fractionated lesions to the thalamic task hubs show different contributions in Figure 5?

3. The least compelling set of results (though not necessarily wrong) is the thalamic prediction of cortical activations. This is because the functional connectivity (FC) matrix used to link the thalamus and cortex was derived from the same data after regressing out task-related variance. However, this process might not be clean enough. The authors do not provide enough details on how task-evoked responses were regressed and how the residuals were assessed to be clean. Perhaps these details could quell my concerns. That said, if available, a stronger test would utilize an FC matrix derived from independent data. Alternatively, I suggest using the FC matrix from dataset 2 to examine cortical projections of dataset 1 (and vice versa).

4. Throughout the manuscript, there is a general dependence on qualitative comparisons instead of quantifying similarities between findings. For example, (a) the spatial similarity of task hubs across the two datasets was not assessed (Figure 1d) (b) similarities between thalamic task hub projections to the cortex (Figure 2b). Please quantify any comparison between the two datasets.

5. This is optional but I believe it would serve the study well if they link their thalamic task hubs to underlying thalamic nuclei using one of the many existing atlases (e.g. Najdenovska et al. nature scientific data 2018 https://www.nature.com/articles/sdata2018270). Even if the links with nuclei is at a coarse level, it could serve as a nice anatomical foundation for future explorations.

---

## [Author Response]

Essential revisions:1. The measure of "task hub" properties that is central to the paper would need to be much better explained and justified. You motivate the measure to be designed to find voxels that are "more flexibly recruited by multiple thalamic activity components", but it is not clear to me at this point that the measure defined on line 634 does this. First, sum_n w_i^2^ is constrained to be the variance of the voxel across tasks, correct? Would sum_n abs(w) be higher when the weights are distributed across components? Given that each w is weighted by the variance (eigenvalue) of the component across the thalamus, would the score not be maximal if the voxel only loaded on the most important eigenvector, rather than being involved in a number of components? Also, the measure is clearly not rotational invariant – so would this result change after some rotation PCA solution? Some toy examples and further demonstrations that show why this measure makes sense (and what it really captures) would be essential. The same holds for the participation index for the resting state analysis.

We thank the reviewer for raising these questions and would like to make a few clarifications. First, the *V* term in the original task hub definition refers to the percentage (%) of variance explained in evoked activity across tasks by a given component, not the variance within the voxel. We have renamed the variance explained term as *EV* to avoid this confusion. Second, we have performed a control analysis to determine factors that contribute to the task hub metric. Specifically, we compared the following two metrics:

The original task hub metric *WxEV* = ∑i=1n|Wi|×EVi

A revised task hub metric *CompW* = ∑i=1n|Wi|

The difference between these two metrics is that the revised metric is not weighted by the percentage variance explained of each component (*EV*), so the resulting spatial pattern should only reflect the “sum of component weights (sum(abs(w))).” We found the spatial map of *CompW* to be remarkably similar to the original metric *WxEV* (spatial correlation was 0.95 for the MDTB dataset and 0.86 for the N and N dataset). This suggests that the *EV* (variance explained) term had relatively minor contribution to the spatial topography when compared to the abs(W) term. In other words, the component weights more strongly contributed to the observed maps (Author response image 1).

**Author response image 1. sa2fig1:** 

In addition, we recalculated *CompW* using lower ranked components (components with lower *EV*, total summed variance <2%, components #11^th^ to 20^th^). The result revealed a different spatial topography, suggesting that top components with higher explained variances do have a distinct spatial topography (spatial correlation for MDTB = -0.03, for N and N = 0.02). We have included this additional plot in Figure 2-supplement 1.Given these findings, for the revised manuscript we now only use the *CompW* metric as the hub metric, and we updated its definition with the following text in the revised manuscript:

“We reasoned that if a thalamic voxel is broadly participating in multiple tasks, it will express stronger weights for top thalamic activity components that explained a large amount of variance in evoked activity across tasks. Therefore, to further map “task hub” regions in the human thalamus, we calculated a metric by summing each voxel’s absolute component weight for the top ten components.” (line 165)

This revised description is a more accurate description of what the metric is measuring, and we have updated the language throughout the manuscript.

As suggested by the reviewer, we have also included a supplemental figure to describe the intuition of the participation coefficient (PC) metric for measuring network hubs (Figure 2—figure supplement 2). We also provided additional details on this measurement in the methods section (line 643). Briefly, PC is a measure of the weight of between-network connectivity (estimated from functional connectivity) for each thalamus voxel, normalized by their expected functional connectivity weight:

PC=1− ∑s=1N(KisKi)2 where Kis is the total functional connectivity weight between thalamus voxel *i* and cortical ROIs within network *s*, and Ki is the sum of functional connectivity weight between voxel *i* and all cortical ROIs. *N* is the total number of cortical networks. If a voxel has connectivity uniformly distributed to all cortical networks, then its PC value will be close to 1; on the other hand, if connectivity is concentrated to a specific network, its PC value will be close to 0. Note that it is necessary to threshold the FC matrix for this calculation to remove negative functional connectivity weights. In our analyses, we calculated PC by thresholding the thalamocortical functional connectivity to preserve the top 1% to top 15% of functional connectivity weights. The results were then averaged across thresholds. As suggested by the reviewer, a toy example is how provided in Figure 2—figure supplement 2.

For the example on the left of Figure 2—figure supplement 2, the black ROI only has connectivity concentrated with ROIs in network B, therefore the ∑s=1N(KisKi)2 term will be 1, resulting in a PC value of 0. For the example on the right, its connectivity is distributed to both network A and B, therefore the ∑s=1N(KisKi)2 term will be 0.5, resulting in a higher PC value of 0.5

2. The least compelling set of results (though not necessarily wrong) is the thalamic prediction of cortical activations. This is because the functional connectivity (FC) matrix used to link the thalamus and cortex was derived from the same data after regressing out task-related variance. However, this process might not be clean enough. The authors do not provide enough details on how task-evoked responses were regressed and how the residuals were assessed to be clean. A stronger test would utilize an FC matrix derived from independent data. Alternatively, I suggest using the FC matrix from dataset 2 to examine cortical projections of dataset 1 (and vice versa).

We agree that it is important to verify whether task-evoked activations have been removed from the timeseries data prior to calculating the FC matrix. We ran a control analysis to verify this. Specifically, we correlated the task regressor (stimulus timing convolved with the γ response function) with the thalamus and cortical timeseries data that we used for the activity flow analyses. We found the correlations to be very close to zero (correlation with cortical ROI timeseries, mean r = -2.02 * 10^-10^, SD = 1.06* 10^-8^; correlation with thalamic voxel timeseries, mean r = 4.37 * 10^-10^, SD = 1.27* 10^-10^). These weak correlations suggest that task-evoked activations have been successfully cleaned prior to FC calculations, and thus our activity flow analysis is likely not biased by the shared variance between task-evoked activations and timeseries used for calculating the FC matrix. We have included this information on line 679.

3. Throughout the manuscript, there is a general dependence on qualitative comparisons instead of quantifying similarities between findings. For example, (a) the spatial similarity of task hubs across the two datasets was not assessed (Figure 1d) (b) similarities between thalamic task hub projections to the cortex (Figure 2b). The comparison between the two datasets should be better quantified.

We appreciate the reviewer for pointing this out. We now report the following spatial correlations in the revised manuscript.

– The spatial correlation between task hubs from the two datasets across 2445 thalamus voxels was r = 0.36 (Figure 2A, line 174).

– The spatial correlation between FC hub (PC) from the two datasets was r = 0.55 (Figure 2A, Line 193).

– The spatial correlation of cortical projection of task hub between the two datasets is r = 0.46 (Figure 2B, Line 200).

– The spatial correlation between effects of simulated lesions on activity flow predictions between the two datasets was r = 0.29 (Figure 5B, Line 331).

Please note, we only performed statistical test on the spatial correlation between task hub and FC hub, as that was the only inferential statement we made in the manuscript (Line 191). We note that all other observed spatial correlations were stronger than the spatial correlations between FC and task hubs.

4. For the activity flow analysis, the null models (which need to be explained better) appear weak (i.e. no differences across tasks?), and it is no small wonder that the thalamus does significantly better. The Pearson correlations are not overwhelmingly impressive either. To give the reader a feel for how good/bad the prediction actually is, it would be essential that the authors would report noise ceilings – i.e. based on the reliability of the cortical activity patterns and thalamic activity patterns, what correlation would the best model achieve (see King et al., 2022, BioRxiv, as an example).

We thank the reviewer for pointing out this limitation. Following methods proposed in [39], we calculated the noise ceiling for each region, task, model, and subject, using split-half reliability estimates from the activity flow model (Modeled_Actx) and observed cortical responses (Observed_Actx):Noise Ceiling = Reliability(Modeled_Actx)× Reliability(Observed_Actx)
Reliability(Modeled_Actx) is the split half reliability of activity flow prediction from the source ROI, and Reliability(Observed_Actx) is the split half reliability of the observed cortical evoked responses. Details on how noise ceiling estimates were calculated are now reported in line 703. The noise ceiling level is reported in the manuscript (Figure 4—figure supplement 1), and results in figures 4 and 5 are updated with the noise normalized value (divided by the noise ceiling).

We also included a third null model, as suggested by the reviewer. This null model averaged the evoked response amplitudes across all conditions for every voxel, which assumed there is no difference in the voxel-wise evoked response patterns between conditions (line 254). Furthermore, as requested by the reviewer, we included additional details to better explain the assumption of each null model:

“The first null model randomly shuffled thalamic evoked responses (the “null model”), which assumed no spatial structure in thalamic task-evoked responses. The second null model set all thalamic evoked responses to the same value (“1”) across all voxels (the “uniformed” evoked model”), which assumed that cortical activity patterns are determined only by the summated inputs from thalamocortical FC patterns. The third null model was constructed by averaging the evoked response amplitudes across tasks for every thalamic voxel, which assumed that there is no difference in voxel-wise thalamic activity patterns between tasks.”

While the raw Pearson correlation values from the activity flow model were weaker than some of the previous activity flow studies (i.e., Cole et al., 2016), we would like to note that there is one important difference between our approach and previous studies. Previous studies included “multiple” cortical regions to build the activity flow prediction model, whereas our study tested a different question to evaluate the predictive power of a “single” subcortical region, the thalamus. We found that after accounting for the noise ceiling, thalamic activity alone was able to predict approximately 38% of the systematic variance across 400 cortical ROI activity patterns.

Reviewer #2 (Recommendations for the authors):The findings presented here are important. The following recommendations would make the conclusions stronger:1. The study would benefit from linking the thalamic task hubs to canonical resting-state networks defined in the thalamus (e.g. Ji et al. 2019 NeuroImage). Do task hubs mainly overlap with one functional network [e.g. the frontoparietal network (FPN)] or do they cross multiple functionally distinct networks? The latter would suggest some fractionation of the identified hubs (which is alluded to in the current results, due to the non-replication of the posterior thalamic task hub across datasets). It would be most interesting if task hubs ended up occupying voxels at the intersection of multiple RSNs (similar to how Power et al. 2013 Neuron defined hubs).

We thank the reviewer for raising this question. We do not think that task hub in the thalamus is a monolithic structure, instead it is biologically more plausible that different subdivisions within the thalamus, either defined histologically or functionally, can exhibit strong task hub properties. This interpretation is more in line with our previous studies demonstrating that network hubs are distributed across nuclei and functional parcellations within the human thalamus (Hwang et al., 2017 JN; Hwang et al. 2021 *eLife*).

To further describe the anatomical distribution of thalamic task hubs, we summarized the spatial distribution of *compW* using two thalamic atlases: (1) The Morel atlas (Krauth et al., 2010 NeuroImage), which defined thalamic nuclei using histological information from postmortem human brains, and (2) a functional network parcellation atlas developed by Thomas Yeo (https://twitter.com/bttyeo/status/1248992927830781952), which defined thalamic parcels based on its thalamocortical functional connectivity with the 7 canonical resting cortical networks that we used for the network hub analysis (Figure 2A). We excluded the “limbic network” parcellation from this analysis because only 4 thalamic voxels were assigned to it. These results are presented in the new Figure 3.

In both datasets, we found the task hub value to be higher in the anterior and mediodorsal nuclei. These two thalamic nuclei are known to have strong reciprocal connectivity with the medial frontal and lateral frontal regions. We also found the medial pulvinar to show high task hub value but only in the MDTB dataset, likely because the N and N dataset did not show high task hub value in the posterior thalamus.

For the functional network parcellation atlas, between datasets discrepancies were also found in posterior thalamus. For example, in the MDTB dataset, the DF parcellation, which overlapped with the anterior medial, medial, and posterior medial thalamus, was found to show the highest task hub value. For the N and N dataset, the posterior thalamus did not exhibit high task hub value, but the FP parcellation, which covered the dorsal bank of the medial thalamus, showed the highest task hub value. Overall, these results suggest that thalamic task hubs do not mainly overlap with one functional network parcellation or thalamic nucleus, instead cover multiple parcellations/nuclei.

Krauth, A., Blanc, R., Poveda, A., Jeanmonod, D., Morel, A., and Székely, G. (2010). A mean three-dimensional atlas of the human thalamus: generation from multiple histological data. *Neuroimage*, *49*(3), 2053-2062.

Schaefer, A., Kong, R., Gordon, E. M., Laumann, T. O., Zuo, X. N., Holmes, A. J., … and Yeo, B. T. (2018). Local-global parcellation of the human cerebral cortex from intrinsic functional connectivity MRI. *Cerebral cortex*, *28*(9), 3095-3114.

Reber, J., Hwang, K., Bowren, M., Bruss, J., Mukherjee, P., Tranel, D., and Boes, A. D. (2021). Cognitive impairment after focal brain lesions is better predicted by damage to structural than functional network hubs. *Proceedings of the National Academy of Sciences*, *118*(19), e2018784118.

2. Could fractionating the thalamic task hubs reveal different cortical contributions? The cortical results in Figure 2B are actually not that similar across the datasets (e.g. ventro-medial frontal and posterior cingulate areas). Further, the cortical areas identified occupy much of the association cortices, which is inconsistent with more localized cortical hubs that the authors reference (Bertolero et al. as well as other studies). The posterior portion of the thalamic task hubs is already one potential contributing factor to these differences. There is also evidence that anterior thalamic regions are the most connected to a localized core subset of the FPN (Assem et al. 2020 Cerebral Cortex). Similarly, would simulating fractionated lesions to the thalamic task hubs show different contributions in Figure 5?

Given that we found that task hubs are spatially distributed across different thalamic sub-regions, it is more likely that different subregions will have different coupling patterns with cortical regions. This is perhaps to be expected, especially given the known anatomical thalamocortical projection profile of different thalamic nuclei. We demonstrated this point by performing a control analysis, where we projected the task hub estimates (*compW*) from the following three thalamic nuclei (anterior nucleus [AN], mediodorsal nucleus [MD], and medial pulvinar [PuM]) to the cortex via the activity flow mapping procedure. The results are presented in the Author response imahe 2. As expected, the anterior nucleus showed strong coupling with medial and lateral frontal regions, the mediodorsal thalamus with the lateral frontal cortex and the insula, and the medial pulvinar with the occipitotemporal cortex for the MDTB dataset (less clear pattern for the N and N dataset). Given these are expected from the known thalamocortical anatomy, we opted to not include this control analysis in the main manuscript.

We agree with the reviewer that the frontoparietal association areas found to show strong coupling with thalamic task hubs were more extensive than the more selective task-flexible regions found in previous papers. We have revised the Discussion section to make it clear that our results showed thalamic coupling with multiple regions across the broad frontoparietal cortices, and among the overlapped regions some selective frontoparietal regions have been shown to be part of a flexible multiple-demand system (line 491).We also simulated lesions to different thalamic nuclei that we found to exhibit high task hub values. Specifically, we simulated lesions separately for the anterior nucleus (AN), the mediodorsal nucleus (MD), and the medial pulvinar (PuM), which corresponded to the anterior, medial, and posterior medial portion of the thalamus that we found to show strong task hub values. We found that simulated lesions to these nuclei significantly impaired activity flow model performance (MDTB: AN mean reduction = 17.33%, SE = 5.71%, MD mean reduction = 16.98%, SE = 6.06%, PuM mean reduction = 17.45%, SE = 6.42%; N and N AN: mean reduction = 13.59%, SE = 9.24%, MD: mean reduction = 14.29%, SE=8.83%, PuM mean reduction = 6.7%, SE = 9.26%), with the exception that weaker reduction in activity flow model performance was observed for PuM in the N and N dataset. Critically, we did not observe a statistically significant difference in the effects of simulating lesions to these nuclei. Thus, the main thalamic task hub regions in the anterior, medial, and posterior medial thalamus appear to have similar contributions to activity flow model prediction. Given that we did not observe significant differences, we opted to not include these control analyses to the main manuscript.

3. The least compelling set of results (though not necessarily wrong) is the thalamic prediction of cortical activations. This is because the functional connectivity (FC) matrix used to link the thalamus and cortex was derived from the same data after regressing out task-related variance. However, this process might not be clean enough. The authors do not provide enough details on how task-evoked responses were regressed and how the residuals were assessed to be clean. Perhaps these details could quell my concerns. That said, if available, a stronger test would utilize an FC matrix derived from independent data. Alternatively, I suggest using the FC matrix from dataset 2 to examine cortical projections of dataset 1 (and vice versa).

Please see our response to essential revisions point #2.

4. Throughout the manuscript, there is a general dependence on qualitative comparisons instead of quantifying similarities between findings. For example, (a) the spatial similarity of task hubs across the two datasets was not assessed (Figure 1d) (b) similarities between thalamic task hub projections to the cortex (Figure 2b). Please quantify any comparison between the two datasets.

Please see our response to essential revisions point #3.

5. This is optional but I believe it would serve the study well if they link their thalamic task hubs to underlying thalamic nuclei using one of the many existing atlases (e.g. Najdenovska et al. nature scientific data 2018 https://www.nature.com/articles/sdata2018270). Even if the links with nuclei is at a coarse level, it could serve as a nice anatomical foundation for future explorations.

Please see our response to reviewer 2 comment #1. We have included this suggested analysis in the new figure 3.